# Impact of the Co-Administration of N-3 Fatty Acids and Olive Oil Components in Preclinical Nonalcoholic Fatty Liver Disease Models: A Mechanistic View

**DOI:** 10.3390/nu12020499

**Published:** 2020-02-15

**Authors:** Rodrigo Valenzuela, Luis A. Videla

**Affiliations:** 1Nutrition Department, Faculty of Medicine, University of Chile, Santiago 8380000, Chile; 2Nutritional Sciences Department, Faculty of Medicine, University of Toronto, Toronto, ON M2J4A6, Canada; 3Molecular and Clinical Pharmacology Program, Institute of Biomedical Sciences, Faculty of Medicine, University of Chile, Santiago 8380453, Chile; lvidela@med.uchile.cl

**Keywords:** extra virgin olive oil, hydroxytyrosol, n-3 polyunsaturated fatty acids, nonalcoholic fatty liver disease

## Abstract

Nonalcoholic fatty liver disease (NAFLD) is present in approximately 25% of the population worldwide. It is characterized by the accumulation of triacylglycerol in the liver, which can progress to steatohepatitis with different degrees of fibrosis, stages that lack approved pharmacological therapies and represent an indication for liver transplantation with consistently increasing frequency. In view that hepatic steatosis is a reversible condition, effective strategies preventing disease progression were addressed using combinations of natural products in the preclinical high-fat diet (HFD) protocol (60% of fat for 12 weeks). Among them, eicosapentaenoic acid (C20:5n-3, EPA) and docosahexaenoic acid (C22:5n-3, DHA), DHA and extra virgin olive oil (EVOO), or EPA plus hydroxytyrosol (HT) attained 66% to 83% diminution in HFD-induced steatosis, with the concomitant inhibition of the proinflammatory state associated with steatosis. These supplementations trigger different molecular mechanisms that modify antioxidant, antisteatotic, and anti-inflammatory responses, and in the case of DHA and HT co-administration, prevent NAFLD. It is concluded that future studies in NAFLD patients using combined supplementations such as DHA plus HT are warranted to prevent liver steatosis, thus avoiding its progression into more unmanageable stages of the disease.

## 1. Introduction

Nonalcoholic fatty liver disease (NAFLD) is considered the hepatic expression of the metabolic syndrome, which is characterized by (i) the accumulation of triacylglycerol (TG) in the cytoplasm of hepatocytes; (ii) a 25% prevalence in the population worldwide; (iii) the manifestation of a spectrum of liver alterations including simple hepatic steatosis, steatosis with inflammation (steatohepatitis, NASH), and different degrees of fibrosis; (iv) the absence of approved pharmacological therapies; and (v) the estimation that 20%–30% of NAFLD patients may progress to NASH [1]. The latter inflammatory condition is estimated to evolve to cirrhosis in 7%–25% of the patients [1,2] and represents an indication for liver transplantation with a consistently increasing frequency [3]. In this scenario, the diminution in the energy intake with concomitant physical activity is the first strategy for the adequate handling of NAFLD. However, when these lifestyle changes are inefficient, pharmacological management, dietary interventions, or the use of different therapeutic agents has become the second line of attack [4].

Considering that hepatic steatosis is a reversible condition, effective therapies are required to avoid the progression of steatosis to NASH and fibrosis, otherwise causing chronic liver disease involving irreversible modifications [5,6]. Furthermore, NAFLD is a multifactorial illness that includes complex metabolic changes, not only in the liver but also in adipose tissue and muscle, which suggest that monotherapies are unlikely to elicit successful responses [4]. In view of these considerations, new therapeutic approaches were evaluated using the combination of bioactive compounds, a strategy that is characterized by (i) the use of lower doses of compounds than monotherapies with shorter supplementation periods to minimize possible side effects; and (ii) the involvement of compounds with protective effects that are exerted through different or similar mechanisms of action, thus allowing synergistic or additive actions and a more efficient control of the damaging effects [4,7]. For example, (i) preservation of liver tissue regeneration post-hepatectomy can be obtained by a L-3,3′,5-triiodothyronine (T_3_) plus methylprednisolone treatment [8]; (ii) high-fat diet (HFD)-induced liver steatosis can be diminished by n-3 long-chain polyunsaturated fatty acids (n-3 LCPUFA) and extra virgin olive oil (EVOO) [9]; (iii) combined T_3_ and fish oil supplementation suppresses ischemia-reperfusion inflammatory liver injury [10]; whereas (iv) resveratrol and enalapril improved glucose and lipid profiles by decreasing lipogenic gene expression [11]. Interestingly, an inverse correlation between serum free thyroxine (T_4_) levels and hepatic steatosis was established in overweight and obese patients [12] or with elevated serum thyroid-stimulating hormone concentrations in overweight/obese children [13], while higher baseline levels of T_3_ and T_4_ predict more weight loss, but not weigh regain, in overweight/obese patients with normal thyroid function subjected to weight loss diets [14]. Collectively, these evidences point to the concept that natural products, drugs, and thyroid hormones constitute hormetic agents or hormetins, which favor beneficial effects by acting at low dosages through one or more pathways of maintenance and repair, thus conferring resistance to subsequent, otherwise harmful, conditions of increased stress [15], including metabolic stress [16]. The aim of this article is to discuss recent data concerning the use of natural product co-administration in the prevention of NAFLD development.

## 2. Material and Methods

The review includes several literature searches that considered the metabolic and beneficial effects of n-3 LCPUFA, particularly eicosapentaenoic acid (C20:5n-3, EPA) and/or docosahexaenoic acid (C22:6n-3, DHA), EVOO, hydroxytyrosol (HT), n-3 LCPUFA plus EVOO, or n-3 LCPUFA plus HT using in vivo and in vitro models. Study searches were performed using the PubMed database from the National Library of Medicine—National Institutes of Health. Emphasis was placed on the participation of n-3 LCPUFA, EVOO, and HT as potential hepatoprotective compounds in liver steatosis models.

## 3. Influence of Energy Intake and Diet Composition on Liver Steatosis Development

Diet has a relevant role in the development and progression of NAFLD, since a high energy intake and consumption of specific nutrients have a direct impact on the abnormal accumulation of TG in the liver, a hallmark of NAFLD [4]. High intake of nutrients that include saturated fatty acids (FA) such as palmitic acid (C16:0) [17,18,19] and *trans* FA of industrial origin [20] decrease FA oxidation (FAO), stimulate the synthesis and secretion of TGs, and trigger lipotoxic effects in the liver [4,17,18]. Moreover, high intake of n-6 PUFA, especially linoleic acid (C18:2n-6), and low consumption of n-3 LCPUFA (EPA and DHA) also appear to favor the development of hepatic steatosis [21]. Furthermore, the consumption of fructose has significantly increased worldwide with the growth of processed foods using high fructose corn syrup [22]. In the liver, fructose metabolism is different from glucose metabolism and proceeds without regulation, thus providing excess acetyl units that promotes a hepatic prolipogenic state [23], with further ATP depletion, oxidative stress, n-3 LCPUFA depletion, and development of a proinflammatory state [24,25]. Moreover, derangements in liver iron and copper homeostasis are related to the development of NAFLD. An increase in liver iron levels is associated with advanced lesions in patients with NAFLD [26], a condition in which the levels of hepatic n- 3 LCPUFA are diminished in relation to the development of oxidative stress, triggering de novo lipogenesis over FAO [27]. Unlike iron, the content of hepatic copper is decreased in NAFLD patients, a situation that favors TGs and cholesterol biosynthesis [28,29], with the concomitant oxidative stress enhancement due to diminution in the antioxidant potential of the liver and the induction of iron overload [30].

## 4. Diminution of Liver Steatosis by Natural Products Co-Administration

Mice subjected to HFDs comprising 60% of the total calories as saturated fat, mainly from lard, for 12 weeks is considered as a suitable experimental approach for liver steatosis development, similar to that found in NAFLD patients [31]. Under these conditions, fatty liver with a steatosis score of around 2 is developed (Figure 1A,B), which corresponds to 33% to 66% of hepatocytes infiltrated with fat [32]. Concomitantly, HFD did not alter serum transaminase levels or induce overt hepatic inflammatory hallmarks; however, liver oxidative stress and inflammatory cytokine expression were significantly enhanced, thus inducing a proinflammatory state [33]. HFD-induced liver steatosis is diminished by 66% to 83% with EPA plus DHA [34,35,36], DHA plus EVOO [37], or EPA plus hydroxytyrosol (HT) supplementations (Figure 1C) [38,39,40]. The attenuation of HFD-triggered hepatic steatosis by these combinations is comparable to the sum of effects elicited by the separate supplementations, thus reaching additive responses [37,38,39,40].

In the case of the co-administration of EPA and DHA, the partial anti-steatotic effects are elicited either when the supplementation is carried out along with the HFD for 12 weeks [33,35] (Figure 1C) or when animals given HFD for 12 weeks are subjected to a control diet plus EPA and DHA for 8 additional weeks [35]. These findings indicate that EPA + DHA supplementation partially prevents fatty liver development, an effect that is associated with different molecular mechanisms triggered by the n-3 LCPUFA (Figure 2). EPA is a precursor of DHA [41], which contributes to the enhancement in liver DHA availability [33]. Compared with EPA, DHA exhibits a greater chemical reactivity [7] associated with the formation of active derivatives (Figure 2) [42,43,44] and affords more beneficial effects than EPA [45,46]. DHA binding to peroxisome proliferator-activated receptor-α (PPAR-α) leads to PPAR-α activation with enhanced binding capacity to DNA, promoting the expression of genes encoding for proteins involved in different aspects of FA metabolism [47]. In the liver, these include FA uptake through membranes, intracellular trafficking and activation, FAO, and phospholipid remodeling [48]. FAO is associated with upregulation of the hepatic energy sensing cascade involving fibroblast growth factor 21 (FGF21), AMPK-activated protein kinase (AMPK), and PPAR-γ coactivator-1α (PGC-1α) (Figure 2) [40], as PPAR-α also controls FGF21 expression [48]. Furthermore, elevated liver FAO is likely to be subsidized by the increased transcription of the carnitine/acylcarnitine carrier (CAC) gene elicited by n-3 LCPUFA (Figure 2), a component of the carnitine cycle catalyzing the transport of fatty acyl units into mitochondria to undergo oxidation [49]. Importantly, a recent report identified 25 genes that are dysregulated during steatosis progression to NASH, including the significant loss of those encoding for PPAR-α and PGC-1α, which drastically disturb mitochondrial function [50]. Furthermore, progression of steatosis to NASH decreases the content of liver LCPUFA by 59%, 78%, and 89% compared with control values in mice subjected to a Western diet (40% energy as fat) for 4, 10, and 24 weeks, respectively [51]. Besides supporting liver FAO, n-3 LCPUFA promote the decline of hepatic de novo lipogenesis through at least three mechanisms of action, namely, (i) diminution of the nuclear availability of the lipogenic transcription factor sterol regulatory element binding protein-1c (SREBP-1c) via AMPK-mediated serine-365 phosphorylation of nascent SREBP-1c, resulting in inhibition of the intramembrane proteolysis of the nascent SREBP-1c (Figure 2; [52]); (ii) DHA-dependent downregulation of the expression of SREBP-1c and target lipogenic enzymes through interaction with G-protein-coupled receptor 40 (GPR40) [53]; and (iii) n-3 LCPUFA-dependent repression of the citrate carrier (CIC) expression secondary to SREBP-1c downregulation, thus decreasing the transport of mitochondrial acetyl-CoA units as citrate outside mitochondria for FA synthesis (Figure 2) [49]. These findings support the contention that HFD-induced hepatic steatosis is partly decreased by n-3 LCPUFA in association with a change in the prolipogenic pattern of the liver imposed by HFD, through the establishment of a significantly lower SREBP-1c/PPAR-α ratio [54]. Notably, liver SREBP-1c upregulation, PPAR-α downregulation, and depletion of n-3 LCPUFA are also observed in obese patients with NAFLD, the respective SREBP-1c/PPAR-α ratios being inversely correlated with the n-3 LCPUFA levels and directly associated with insulin resistance [55].

Finally, the antisteatotic effect of n-3 LCPUFA supplementation is also related to their high susceptibility to undergo spontaneous lipid peroxidation with formation of J3-isoprostanes, which promote the activation of transcription factor nuclear factor erythroid 2-related factor 2 (Nrf2) [44]. Nrf2 activation triggers antioxidant defenses against the oxidative stress prevailing NAFLD [33,57], which may avoid (i) further n-3 LCPUFA depletion that favors steatosis development; and (ii) the induction of endoplasmic reticulum (ER) stress through protein oxidation/unfolding that upregulates lipogenic SREBP-1c and PPAR-γ expression [58]. Supporting the importance of the actions of n-3 LCPUFA on hepatic steatosis development, alterations in gut microbiota from fat-1 mice protect the liver against high-fat/high-sucrose diet-induced NAFLD [59]. These fat-1 transgenic mice encode an n-3 FA desaturase that converts n-6 to n-3 LCPUFA, thus endogenously increasing the levels of n-3 LCPUFA [59] and their antisteatotic signaling (Figure 2). In this respect, n-3 LCPUFA are considered as prebiotics influencing the composition of gut microbiota, which is altered following HFD-feeding (dysbiosis), thus representing potential agents able to restore eubiosis in the intestinal flora that may abrogate the pathological changes induced by HFD [60].

Besides, the antioxidant effects of n-3 LCPUFA allow the abrogation of the redox activation of nuclear factor-κB (NF-κB) that promotes inflammation expansion [61]. N-3 LCPUFA-induced anti-inflammatory responses are linked to the production of several oxidation products including EPA- and DHA-derived E-series and D-series of resolvins, DHA-derived D-protectins, and epoxygenated FA, acting as NF-κB downregulators (Figure 2) [62]. In this respect, EPA supplementation increases the hepatic levels of EPA and DHA, resulting in increased levels of resolvins RvE1/2 and RvD1/2 [39], whereas DHA elicited similar results in RvD1/2 availability [63]. The activation of NF-κB with induction of the inflammasome NOD-like receptor protein 3 (NLRP3) components is also a target of the n-3 LCPUFA-dependent actions limiting inflammation, which may involve (i) interference of NF-κB activation by activated PPAR-α [64]; (ii) inhibition of NLRP3 activation [65,66]; and (iii) a DHA-G-protein receptor 120 (GPR120) interaction [66] (Figure 2).

In addition to EPA and DHA co-administration, the combined DHA and EVOO [37] or EPA plus HT [37,38,39,40] administration also elicited a diminution of HFD-induced liver steatosis. The study using DHA plus EVOO revealed that HFD induced fat accumulation in more than 60% of the hepatocytes with 160% increase in the content of TG, both of which were diminished by 10%–20% and 47%, respectively, by DHA plus EVOO [37]. Mechanistically, the partial antisteatotic effect of DHA plus EVOO mainly relies on the processes set in by DHA (Figure 2), with EVOO having a secondary role at the dosages used [37]. Nonetheless, EVOO oil improves the postprandial glycemic and lipid profiles in patients with impaired fasting glucose [67]. Additionally, acute high-polyphenols EVOO intake is able to modify the transcriptome of peripheral blood mononuclear cells through the modulation of different pathways associated with the pathophysiology of cardio-metabolic disease and cancer [68]. In this context, the beneficial effects of EVOO on the liver are exerted through its major (monounsaturated FA (MUFA) and polyunsaturated FA (PUFA)) and minor (HT and tocopherols) constituents (Figure 3) [69,70,71]. Similarly to n-3 LCPUFA [46], EVOO attenuate diet-induced risks factors for metabolic syndrome, by favoring the relative abundance of prebiotic microbiota [72]. At the dosage used (50 mg/kg/day), EVOO supplementation alone did not modify liver PPAR-α signaling in control and HFD groups, whereas in mice subjected to HFD, it elicited minor changes in parameters related to de novo lipogenesis, oxidative stress, and inflammation [37]. A comparable situation occurs with the combined administration of EPA and HT, in which attenuation in HFD-induced steatosis development is achieved, an effect characterized by the additivity of EPA and HT effects that mainly depend on HT [37,38,39,40]. Although HT exerts beneficial outcomes associated with its significant direct free radical scavenging action [73,74] (Figure 3), alternate mechanisms of action involve (i) antioxidant responses via Nrf2 activation [75]; (ii) antisteatotic effects underlying PPAR-α [76] and FGF21 [77] upregulation and ER stress downregulation; [78] (iii) improvement of mitochondrial oxidative function through AMPK/PGC-1α signaling to favor FAO [40,75,79]; and (iv) prevention or resolution of inflammatory processes due to NF-κB deactivation [76,80].

## 5. Suppression of Liver Steatosis Development by the Co-Administration of Docosahexaenoic Acid and Hydroxytyrosol

The partial antisteatotic effect of the co-administration of natural products during HFD feeding, namely, EPA and DHA, DHA and EVOO, and EPA and HT [33,34,35,36,37,38,39,40], indicate that the threshold required to obtain effective functional responses is not attained. This may reflect that the interaction of the underlying mechanisms (Figure 2 and Figure 3) is inadequately exerted in relation to either the intracellular levels of the agents that are actually achieved or differences in their potency to trigger significant responses. In the case of the supplementation with DHA and HT, however, these factors seem to be overcome, since (i) liver DHA levels are higher than those of EPA under normal conditions [9,37,38,40]; (ii) DHA administration to control and HFD-fed mice reaches higher than basal DHA levels in the liver [37]; (iii) DHA is more reactive [7] and beneficial than EPA [45,46,81]; and (iv) the in vivo HT supplementation represents a greater HT dosage (5 mg/kg/day) [38,39,40] than that given as EVOO (Figure 3), thus inhibiting steatosis development (Figure 1D). Accordingly, under conditions of combined DHA and HT administration, an additive antisteatotic effect is attained, since the hepatic steatosis score induced by HFD is diminished by 64%, 38%, and 100% by DHA, HT, and DHA plus HT supplementation, respectively [56]. The above outcome of DHA plus HT involves the normalization of key hepatic metabolic functions that are deranged by HFD, namely, the upregulation of lipogenic SREBP-1c signaling and the downregulation of the pro-FAO action of PPAR-α, thus leading to basal SREBP-1c/PPAR-α ratios avoiding fat accumulation. Moreover, the pro-inflammatory status induced by HFD is also abrogated in relation to the regularization of the NF-κB signaling, which is reinforced by the significant enhancement in liver resolvin availability [56]. These two actions of the DHA and HT co-administration are associated with the normalization of the HFD-induced oxidative stress status, which is achieved through upregulation of the Nrf2 signaling and direct interception of free radicals (Figure 2 and Figure 3), thus avoiding DHA lipid peroxidation and NF-κB redox activation [56].

## 6. Conclusions

The preclinical studies discussed in this review suggest that the combined supplementation with natural products, including EPA and DHA, DHA and EVOO, or EPA plus HT, has a positive impact on NAFLD by diminishing hepatic fat deposition (Figure 1C) and the development of a proinflammatory state by HFD [34,35,36,37,38,39,40], with DHA and HT co-administration preventing liver steatosis development completely (Figure 1D) [56]. Although interventions with joined DHA and HT in human NAFLD are not available at present time, most, but not all, trials with n-3 LCPUFA show improvement in hepatic lipid deposition in adult and paediatric NAFLD patients within a 1 to 2 year time period, without diminution or exacerbation of NASH [54,82,83], DHA being more potent than EPA regarding the suppression of hepatic lipogenesis [84]. In relation to HT, (i) a study in healthy volunteers revealed that a dose of 15 mg/day for 3 weeks exerted positive effects on human health by diminishing parameters related to oxidative stress, with improvement in lipid and plasma profile [85]; (ii) a combined HT and vitamin E protocol improved hepatic steatosis and oxidative stress in children with NAFLD [86]; whereas (iii) the plasma levels of tissue inhibitor of metalloproteinases 1 (TIMP-1) in the group of patients receiving HT (15 mg/day for 63 days) were significantly lower than those levels found in the control group after the epirubicin-cyclophosphamide chemotherapy [87]. Recently, several meta-analyses established that n-3 LCPUFA supplementation (especially DHA) is useful in the dietary management of patients with NAFLD (adults and children), but additional trials are needed to better understand the effects of n-3 LCPUFA on histological outcomes in patients with NASH [88,89,90,91]. Additionally, new omics techniques have proven the beneficial effects of DHA (alone or co-administrated with other products) on hepatocyte lipidome [92]. These antecedents and the molecular mechanisms discussed for DHA and HT co-administration warrant future studies in NAFLD patients, to attained liver steatosis resolution, thus avoiding its progression into more unmanageable stages of the disease. The combined supplementation with DHA and HT may be also of importance in the prevention of metabolic dysregulations associated with the metabolic syndrome, type-2 diabetes, and cardiovascular disease, along with human autoimmune diseases such as rheumatoid arthritis, systemic lupus erythematosus, multiple sclerosis, and type-1 diabetes, due to the effective preventive effects demonstrated for n-3 LCPUFA in the area [93].

## Figures and Tables

**Figure 1 nutrients-12-00499-f001:**
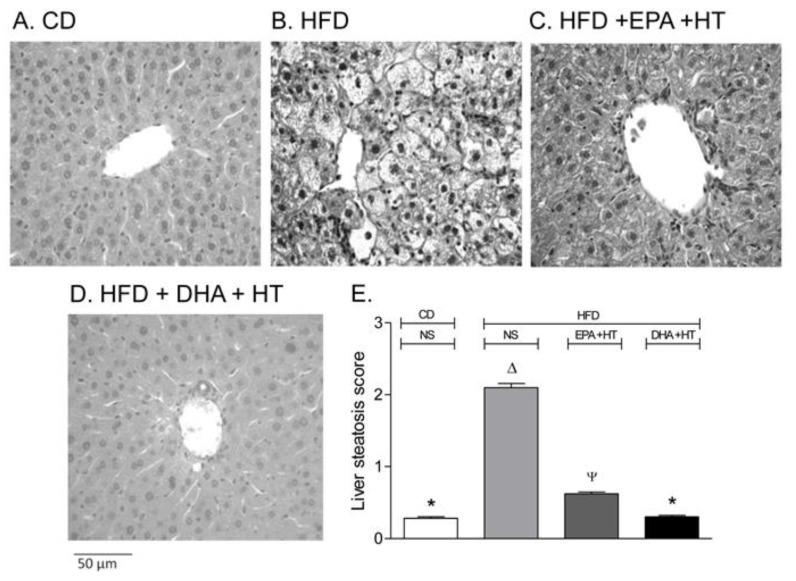
Liver Histological assessment in mice subjected to (**A**) control diet (CD), (**B**) high-fat diet (HFD) without supplementation (NS), (**C**) HFD supplemented with eicosapentaenoic acid (EPA) plus hydroxytyrosol (HT) and (**D**) HFD, supplemented with docosahexaenoic acid (DHA) plus HT. Weaning male C57BL/6J mice (*n* = 7 per experimental group) were allowed free access to a CD (10% fat, 20% protein, and 70% carbohydrate, with a caloric values of 3.85 KcaL/g; Rodent Diet, Product data D12450B and D12492, Research Diet Inc., USA) or HFD (60% fat, 20% protein, and 20% carbohydrate, with a caloric values of 5.24 Kcal/g; Rodent Diet, Product data D12492, Research Diet Inc., USA) for 12 weeks. Animals subjected to CD (not shown) or HFD were simultaneously supplemented with EPA (50 mg/kg/day) plus HT (5 mg/kg/day) or DHA (50 mg/kg/day) plus HT (5 mg/kg/day) through gavage. Liver samples were fixed in phosphate-buffered formalin, embedded in paraffin, stained with haematoxylin-eosin, and analyzed by optical microscopy in blind fashion describing the presence of steatosis, graded according to Brunt et al. [32]. (**E**) Liver steatosis scores (mean ± SEM; *n* = 7). ^a,b,c,d^ Groups sharing the same lyrics are not significantly different among them according to a two‐way ANOVA and Bonferroni’s post-test (*p* < 0.05). Adapted from Echeverría et al. [38] and Soto-Alarcón et al. [56].

**Figure 2 nutrients-12-00499-f002:**
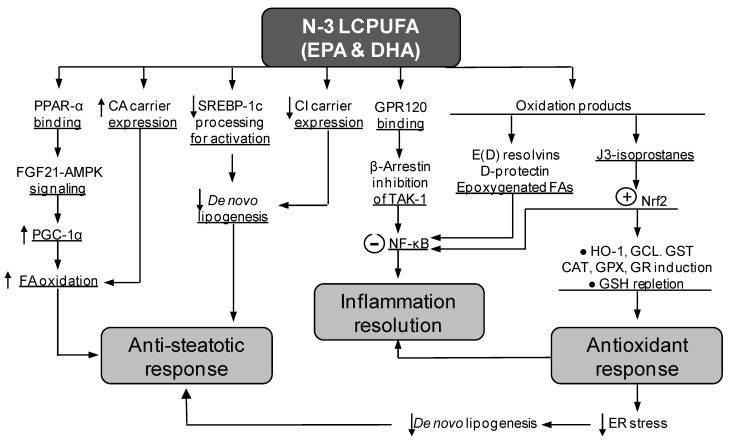
Molecular mechanisms of the n-3 long-chain polyunsaturated fatty acids (n-3 LCPUFA) EPA and DHA explaining antisteatotic and antioxidant responses and inflammation prevention or resolution in the liver. Abbreviations: AMPK, AMP-activated protein kinase; CA carrier, carnitine/acylcarnitine carrier; CAT, catalase; CI carrier, citrate carrier; ER, endoplasmic reticulum; FA, fatty acids; FGF21, fibroblast growth factor 21; GCL, glutamate-cysteine ligase; GPR120, G-protein receptor 120; GPX, glutathione peroxidase; GR, glutathione reductase; GSH, reduced glutathione; GST, glutathione-S-transferase; HO-1, heme oxygenase-1; NF-κB, nuclear factor-κB; Nrf2, nuclear factor erythroid 2-related factor 2; PGC-1α, peroxisome proliferator-activated receptor-γ coactivator-1α; PPAR-α, peroxisome proliferator-activated receptor-α; SREBP-1c, sterol regulatory element binding protein-1c; TAK-1, transforming growth factor-β-activated kinase-1.

**Figure 3 nutrients-12-00499-f003:**
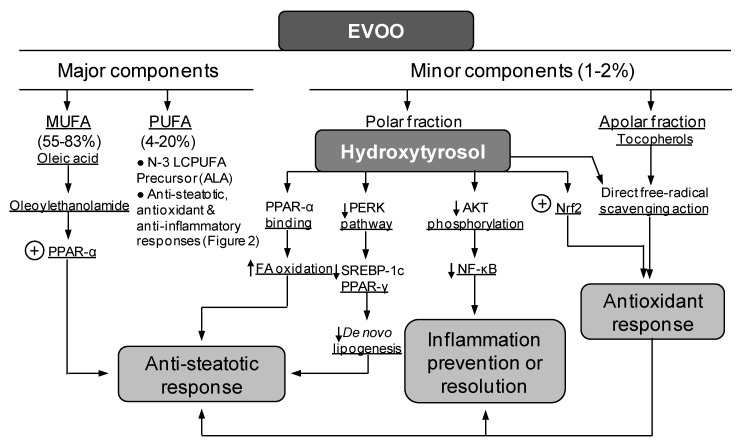
Protective molecular mechanisms associated with extra virgin olive oil (EVOO) in the liver through its major and minor components. Abbreviations: ALA, α-linolenic acid (C18:3n-3); MUFA, mono unsaturated fatty acids; NF-κB, nuclear factor-κB; Nrf2, nuclear factor erythroid 2-related factor 2; PERK, double-stranded RNA-dependent protein kinase (PKR)-like endoplasmic reticulum kinase; PPAR-α(γ), peroxisome proliferator-activated receptor-α(γ); PUFA, polyunsaturated fatty acids; SREBP-1c, sterol regulatory element binding protein-1c.

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
