# Peer review of "Impact of the Co-Administration of N-3 Fatty Acids and Olive Oil Components in Preclinical Nonalcoholic Fatty Liver Disease Models: A Mechanistic View"

_nutrients, 2020, doi:10.3390/nu12020499_

Round 1

Reviewer 1 Report

In the manuscript entitled “Positive Impact of Natural Products Co-Administration on Non-Alcoholic Fatty Liver Disease: A Molecular View”, Valenzuela R., and Videla L., review the impact of polyunsaturated fatty acids EPA and DHA, olive oil and its phenolic compound hydroxytyrosol on NAFLD. They also show some limited original data on a pre-clinical model (mice) of NAFLD when challenged with a high fat diet (60% fat for 12 weeks) that they use to claim mechanism of actions by which the above mentioned compounds exert beneficial effects in mice model. Also, these mechanisms are summarised in figure 2 and 3.

If the topic is in principle of relevance for the NAFLD field as the composition of the diet is a crucial step for MetS and NAFLD development and progression, I am a little bit surprised by the low scientific level of this review:

The manuscript requires a deep revision from a native speaker The authors rely massively on self-citations that is often considered malpractice in science (See https://www.nature.com/articles/d41586-019-02479-7); Despite I see merit in the description of the molecular mechanism of each compound, how they translate into phenotype should be provided as well as a clear discussion of how this translates to human disease (on the basis of available clinical evidence) . As it is, the review is a long list of mechanisms that might be un-useful to most of the potential readers that will not understand it In the title and abstract should be clear that the evidences presented are mainly on pre-clinical/animal models The review is characterised by a very specialised focus on interventions and selected papers. I think a systematic search of the literature should be performed. For example, there are a lot of missing papers that I do not see cited in the manuscript (e.g. PMID: 29844096 in the section 2, or PMID: 27422371 and PMID: 27289163 when discussing EVOO) that address crucial aspects of disease progression and might serve to introduce or discuss the clinical relevance of a specific treatment before discussing molecular mechanisms In keeping with my previous point, there is a significant lack of citations of crucial systematic reviews, metanalyses, and trials that show much limited enthusiasm for the compounds when tested in humans; e.g. multiple studies have shown only mild amelioration of steatosis for EPA and DHA but substantial lack of effectiveness in all the hepatic outcomes; Also, a table with the findings of the different papers (also discussing those that show opposite effects!) should be added. Even though the manuscript is a review, a search strategy for the literature should be added in the method section. This would make the manuscript more scientifically sound. Throughout the manuscript it is not clear as to why authors have put their focus on EPA, DHA, olive oil and hydroxytyrosol. Pre-clinical evidences are emphasised but the references cited are sporadic observations and there is no mention if the results have been replicated by other groups, or if the compounds or the associations have been proven to have relevance in human disease; A paragraph and figure showing the leading mechanisms of NAFLD (increased energy intake, adipose tissue dysfunction and insulin resistance, impaired FAO and mitochondrial function, decreased VLDL export) would have been of help to the reader for a better understanding of the complexity of NAFLD pathophysiology. I like the idea of adding some original data to support a review; However, being these data the backbone of the review: Methods should be clearly explained – especially if these supplements have been fed by gavage or added in the diet Diet product code should be disclosed: if the authors chose to show data, the reader needs to have all the information without the need to read 10 previous papers from the authors to understand how they have been generated … 60% HFD used is a model of really mild NAFLD never progressing toward advanced disease stages (even after 16 months of feeding fibrosis is F0-F1!); in the whole manuscript there is a general overclaim regarding NASH progression that need to be tuned down as the data are in a model of non-progressing fatty liver … the whole review depends from data generated on the wrong model while a Western Diet would have been more appropriate Food intake should be disclosed: one of the reasons of the discrepancies between murine and human data on PUFAs is that the mice eat less if the diet is enriched in PUFA (so they obviously get less weight, are less insulin resistant, and develop less steatosis, hepatic inflammation, and so on) Showing only some example pictures of liver histology does not help to understand if the data are genuine: BW, LW/BW, Lipid profile, ALT at very least – if these data have been diluted in other 10 papers, the authors can seek permission to the journal to reproduce them for a review The title is “a molecular review” but I do not see any molecular data to support any of the mechanisms discussed. So what's the point of adding these histology pictures in a review like this? Conclusions should be more pertinent and clearly explain that the evidences discussed rely mainly to pre-clinical models and that clinical usefulness is either unproven or debated or discounted. Also, the last sentence would have been more appropriate with other metabolic problems such as Metabolic Syndrome, T2D or Cardiovascular Disease …

Minor

Although authors describe the NAFLD spectrum, its definition is somehow spread within the introduction. I would recommend to start the introduction with a sentence where the entire spectrum of NALFD is defined within one sentence to allow the people outside the field to better understand the subject Introduction: NAFLD is not a multifactorial illness that includes complex metabolic changes in the AT and muscle; rather AT/muscle insulin resistance induce metabolic changes promoting NAFLD. Caloric restriction (CR) is both a lifestyle and dietary intervention, please clarify better what you mean in the introduction as it seems that is not a dietary intervention. It is not entirely clear to me why thyroid hormones have been discussed in the introduction. When the “Influence of energy intake and diet composition on liver steatosis development” is described, a more detailed explanation of the mechanisms mentioned would be greatly beneficial to the understanding of the subject. Glucose/ChREBP paradigm, glycaemic index are not cited at all Insulin/SREBP1 paradigm should be better described DNL should be better described Changes in iron and copper metabolism in NASH have been previously described however there is no strong evidence that this is nutritionally led; regarding iron in particular it seems a most-likely hypothesis that could be an alteration of TGFb/BMP signalling causing these changes rather than nutrition Introducing AT expandability, dysfunction and insulin resistance might help to understand the rational why PPARs are beneficial in NAFLD (but I should say gamma2 in particular as only TZDs have been shown to have some effect on  NASH so far) …    It is mentioned a fatty liver score with appropriate reference, yet a definition of the score would be of help to the reader. NAS score from Keiner is more appropriate however ... For instance, oleuropein and tyrosol are also important phenolic compounds that are present in olive oil and can play a role in NAFLD. It is thus important to explain why authors have chosen these compounds and not others (to be clearly discussed in the introduction).

Author Response

Reviewer #1
We thank the reviewer. Your comments were very important to us.

Answers:

We do not consider malpractice in relation to our review article. This is based mainly on our work because there are no studies using combined protocols in the HFD setting by other groups, except for that of Soni et al., (2015)(Ref. 34) using EPA and DHA in a HFD protocol. In addition, as indicated in the Conclusion, “interventions of DHA and HT in human NAFLD are not available at present time”. As suggested by Reviewer-1 and Reviewer-3, the title of the work was re-written in order to indicate preclinical studies and the natural products studied: “Positive Impact of the Co-Administration of Omega-3 Fatty Acids and Olive Oil Components in Preclinical Non-Alcoholic Fatty Liver Disease Models: A Mechanistic View”.

Besides, the aspect of the preclinical nature of the studies reviewed was already indicated in the Abstract of the original version (page 2-lines 26-27).

The review was intended to analyse a very specialized area of research focus on combined protocols either diminishing or avoiding liver steatosis development. In this respect, EVOO was one of the natural products considered, in combination with DHA, the EVOO actions on NAFLD being sufficiently addressed and supported by references 69 and 71. However, we add two more references (both suggested by the reviewer). Reference 65 and 66. In keeping with the previous point, the review was not planned to discuss NAFLD through the numerous systematic reviews and metanalyses available at present time. Since the use of combined protocols in man has not being carried out so far, the Conclusion of our review analyses the results of the interventional studies with n-3 LCPUFAs in adult and paediatric patients with NAFLD or NASH (Refs. 51,80-82), as well as those obtained with HT administration in healthy volunteers (Ref. 83), paediatric NAFLD (ref. 84) or in cancer patients (Ref. 85). These studies provide the dosages of omega-3 and HT used, together with the times of supplementation that could be employed in future combined PUFA-HT studies. But, we added four meta-analyses about the beneficial effects of n-3 LCPUFA in NAFLD (adults and children), and potential use in patients with NASH. Ref 87,88,89 and 90. The focus on EPA, DHA, olive oil and HT was emphasized in the review because they are known to exert beneficial effects at low doses, conferring resistance to the metabolic stress underlying liver steatosis in obesity, as presented in Figures 2 and 3 of the review, mechanisms that supported by previous studies that are properly cited. In particular, mechanisms leading to NAFLD can be found in Refs. 1, 5 and 6. Usually, Methods are not explained in details in review articles. The analysis of HFDs in relation to the content of fat and the period of exposure is available in the work by Lau et al. (2017) (Ref. 29). Besides, HFD-induced fatty liver with a steatosis score over 2 (Figure 1) cannot be considered as a mild NAFLD, since this value indicated that over 66% of the hepatocytes show fat accumulation (Brunt et al., 1999)(ref. 30), which may progress to NASH if the period for treatment is extended to more than 16 weeks As suggested, “a molecular view” was changed to “a mechanistic view” in the title of the review. Regarding saturated fatty acids, we add a new paragraph and a new references. “High intake of nutrients that include saturated FA such as palmitic acid (C16:0) [17,18]”. New reference: 18. Luukkonen, P.K.; Sädevirta, S.; Zhou, Y.; Kayser, B.; Ali, A.; Ahonen, L.; Lallukka, S.; Pelloux, V.; Gaggini, M.; Jian, C.; Hakkarainen, A.; Lundbom, N.; Gylling, H.; Salonen, A.; Orešič, M.; Hyötyläinen, T.; Orho-Melander, M.; Rissanen, A.; Gastaldelli, A.; Clément, K.; Hodson, L.; Yki-Järvinen, H. Saturated Fat Is More Metabolically Harmful for the Human Liver Than Unsaturated Fat or Simple Sugars.Diabetes Care 2018, 41, 1732-39. Respect to the EVOO we added a new paragraph (line 211 to 215) “Nonetheless, EVOO oil improves the post-prandial glycemic and lipid profiles in patients with impaired fasting glucose [65]. Additionally, acute high-polyphenols EVOO intake is able to modify the transcriptome of peripheral blood mononuclear cells through the modulation of different pathways associated with the pathophysiology of cardio-metabolic disease and cancer [66].” and two new references: D'Amore, S.; Vacca, M.; Cariello, M.; Graziano, G.; D'Orazio, A.; Salvia, R.; Sasso, R.C.; Sabbà, C.; Palasciano, G.; Moschetta, A. Genes and miRNA expression signatures in peripheral blood mononuclear cells in healthy subjects and patients with metabolic syndrome after acute intake of extra virgin olive oil. Biochim Biophys Acta 2016, 1861, 1671-80. Carnevale, R.; Loffredo, L.; Del Ben, M.; Angelico, F.; Nocella, C.; Petruccioli, A.; Bartimoccia, S.; Monticolo, R.; Cava, E.; Violi, F. Extra virgin olive oil improves post-prandial glycemic and lipid profile in patients with impaired fasting glucose. Clin Nutr. 2017, 36, 782-87. As also suggested by reviewer-1, the last sentence of the Conclusion section was re-written to: The combined supplementation with DHA and HT may be also of importance in the prevention of metabolic dysregulations associated with the metabolic syndrome, type-2 diabetes and cardiovascular disease, along with human autoimmune diseases such as rheumatoid arthritis, systemic lupus erythematosus, multiple sclerosis and type-1 diabetes, due to the effective preventive effects demonstrated for n-3 LCPUFA in the area [92]. We added a Material and Methods section We correct the editing errors. Additionally, the wording was reviewed by Dr. Raphael Chouinard-Watkins (Canadian native researcher). All changes introduced are indicated in yellow and the references were renumbered.

Thank you for reviewing our manuscript.

Reviewer 2 Report

In the manuscript entitled "Positive Impact of Natural Products Co-Administration on Non-Alcoholic Fatty Liver Disease: A Molecular View", the authors summarized partial current results and understanding how EPA + HT and DHA + HT relieve /reverse hepatic steatosis at the molecular level.  The beneficial effects are achieved by enhancing antioxidant, anti-steatotic and anti-inflammatory responses. I have a few minor concerns regarding this manuscript: 

1) Can EPA + HT and DHA + HT be used as health supplements for prvention even before the occurrence of disease?

2) Is it "Major components" other that "Mayor component" in Figure 3?

Author Response

Reviewer #2

We thank the reviewer. Your comments were very important to us. 

Can EPA + HT and DHA + HT be used as health supplements for prvention even before the occurrence of disease?

Answer:

That would be possible, because the n-3 PUFA (DHA and EPA) and HT. However, it is necessary to conduct these studies in humans.

Is it "Major components" other that "Mayor component" in Figure 3?

Answer:

Error corrected in the figure.

The wording was reviewed by Dr. Raphael Chouinard-Watkins (Canadian native researcher).

All changes introduced are indicated in yellow and the references were renumbered.Thank you for reviewing our manuscript,

Prof. Rodrigo Valenzuela. PhD.

Reviewer 3 Report

In this paper by Valenzuela and Videla, the Authors reviewed the molecular effect of EPA, DHA and component of EVOO in the prevention of NAFLD. The review is clear, elegant and well written.

Minor consideration:

New omics techniques have been used to reveal the effects of DHA (alone or in combination with other natural product) on lipidome in hepatic cells (doi:10.3390/ijms18020359), which use could be applicable and useful also in NAFLD condition. The Authors should consider this possibility.

The use of “natural products” in the title is too generic, in that the focus of the review is on PUFA and components of EVOO.

The paper contains many editing mistake, reference in the text should be place before punctuation and reference list should be rewrite following instruction for Authors

In figure 3 is reported a “table 1” not present in the article

In table 3 change mayor with major

A check by mother tongue is required

Author Response

Reviewer #3

We thank the reviewer. Your comments were very important to us. 

1. New omics techniques have been used to reveal the effects of DHA (alone or in combination with other natural product) on lipidome in hepatic cells (doi:10.3390/ijms18020359), which use could be applicable and useful also in NAFLD condition. The Authors should consider this possibility.

Answer:

New paragraph (line 284 to 286): "Additionally, new omics techniques have proven the beneficial effects of DHA (alone or co-administrated with other products) on hepatocyte lipidome [91]."New references:

[91] V. Ghini, M. Di Nunzio, L. Tenori, V. Valli, F. Danesi, F. Capozzi, C. Luchinat, A. Bordoni. Int J Mol Sci. 2017, 18, E359.

2.  The use of “natural products” in the title is too generic, in that the focus of the review is on PUFA and components of EVOO.

Answer:

We modify the title

New title: "Positive Impact of the Co-Administration of N-3 Fatty Acids and Olive Oil Components in Preclinical Non-Alcoholic Fatty Liver Disease Models: A Mechanistic View"

3. The paper contains many editing mistake, reference in the text should be place before punctuation and reference list should be rewrite following instruction for Authors

Answer:

We correct the editing errors. Additionally, the wording was reviewed by Dr. Raphael Chouinard-Watkins (Canadian native researcher).

In figure 3 is reported a “table 1” not present in the article

Answer:

Error corrected in the manuscript.

In table 3 change mayor with major

Answer:

Error corrected in the figure.

A check by mother tongue is required

Answer: 

The wording was reviewed by Dr. Raphael Chouinard-Watkins (Canadian native researcher).

All changes introduced are indicated in yellow and the references were renumbered.Thank you for reviewing our manuscript,

Prof. Rodrigo Valenzuela. PhD.